# Willingness to Take the Booster Vaccine in a Nationally Representative Sample of Danes

**DOI:** 10.3390/vaccines10030425

**Published:** 2022-03-10

**Authors:** Frederik Juhl Jørgensen, Louise Halberg Nielsen, Michael Bang Petersen

**Affiliations:** Department of Political Science, Aarhus University, 8000 Aarhus, Denmark; lhn@ps.au.dk (L.H.N.); michael@ps.au.dk (M.B.P.)

**Keywords:** booster vaccine willingness, survey research, nationally representative sample

## Abstract

We estimate the willingness to take the booster dose in a representative sample of Danes. We estimate an overall willingness in the adult Danish population of about 87 percent and a willingness of about 95.5 percent among primary vaccine takers. Moreover, we show that these percentages are significantly lower among younger populations, as well as among groups who do not see COVID-19 as a threat to society, those who do not feel that they have the ability to follow recommendations (‘self-efficacy’), those who do not perceive the advice of the health authorities as effective against disease spread (‘response efficacy’), and those who feel that the costs of following recommendations are high (‘response cost’).

## 1. Introduction

The rapid worldwide spread of the Omicron variant of SARS-CoV-2 [1,2] likely occurred due to a combination of the variant’s increased transmissibility and rapidly waning immunity from vaccines against Omicron infection [3,4]. As a consequence, health authorities across the world are currently investing heavily in administering a third dose of vaccination (a booster dose) to increase immunity [3,4]. This strategy poses a fundamental challenge because both epidemic control and epidemic forecasting (including via formal modeling) now hinge on assumptions about human behavior, specifically the proportion of the population that is willing to take the booster dose [5]. The aim of this communication is to provide an estimate of the likely proportion of booster vaccine willingness using Danish data and, in particular, provide novel insights into the main drivers of booster willingness. This may assist the health authorities’ communication efforts to promote booster dose acceptance in Denmark and beyond.

Figure 1 shows the uptake of COVID-19 vaccine boosters per 100 people worldwide [6]. Overall, we observe large variations in booster vaccination rates across countries. Thus, the uptake ranges from between 0 and 10 percent (in countries such as Russia; some South American countries, such as Venezuela; and the African countries for which data are available) to 50+ percent in western European countries generally. Here, we focus on Denmark. With about 62 percent of the total population having received a booster, Denmark is only surpassed by Chile (about 69 percent) in having revaccinated their populations with a third dose of a COVID-19 vaccine. Despite a comparatively high rate of booster acceptance, there is still a substantial part of the Danish population who have not received a booster. For governments’ success in handling the ongoing COVID-19 pandemic, it is key to understand the psychological correlates that motivate citizens’ uptake of boosters, and these psychological correlates will likely be important to motivate citizens outside of Denmark as well. This underlines the research question that we are asking here: what are the main drivers of individuals’ willingness to receive a COVID-19 booster?

The present manuscript identifies and investigates key individual-level psychological correlates of self-reported willingness to take a COVID-19 booster. This analysis is based on large surveys representative of the Danish population (N = 31,721). Building on a general theoretical framework for understanding protective behavior, protective motivation theory, our study provides insights on the correlates of booster willingness and, hence, on the factors that efficient health communication from governments, health authorities, and the media needs to target in order to efficiently promote uptake of boosters among the public now or in future waves of the pandemic.

The framework of protection motivation theory has been shown to be useful for understanding general protective behavior during the COVID-19 pandemic [7,8], as well as understanding vaccine intentions specifically [9,10]. Protection motivation theory stipulates that people’s motivation to comply with risk-relevant recommendations—including taking boosters—is a function of appraisals of feelings of threat and a feeling of being able to cope with the threat through compliance with recommendations [11,12]. Our data therefore capture perceived threats (to society and to individuals) and coping appraisals (perceptions of the ability to follow recommendations, perceptions of the effectiveness of the recommendations, and perceptions of the costs of following recommendations).

## 2. Materials and Methods

### 2.1. Data Collection

Since May 2020, we have been surveilling COVID-19 behavior and its psychological antecedents in daily national representative surveys of Danes. More specifically, we have surveyed approximately 500 (new) respondents each day (by now, the total N includes more than 380,000 individuals). Note that data for this study include only a subset of participants as we focus on booster vaccination willingness that we only started measuring from 15 December 2021 (cf. below). The analysis period for this study is thus 15 December 2021 to 13 February 2022 (N = 31,721). Participants in the survey are Danish citizens aged 18 or older (the total Danish population aged 18 or older amounts to 4,721,691 by the first quarter of 2022 according to data retrieved from Danish Statistics [13]). Participants were recruited using stratified random sampling—on age, sex, and geography—based on the entire database of Danish social security numbers. The sample is thus representative of the Danish population. The response rate was about 25 percent. We did not employ any monetary incentives for participation. Participants were invited to take the survey via eBoks. eBoks is the official electronic mail system used by Danish authorities and other institutions to communicate with citizens. eBoks is linked to an individual’s social security number and follows citizens for their entire life. This link between the social security number and eBoks is the reason that we were able to utilize eBoks as a means of sampling. Note that about 8 percent of the Danish population, mainly older people, are exempt from eBoks. Nonetheless, we observe that the sampled observations are close to the population margins with respect to sex, age, geographic region, and vote choice at the last first-order national election, whereas the sample is biased towards people with higher education and people among the adult population who have been vaccinated (see Table 1). To correct for these biases, we conducted the analyses on the weighted data employing entropy balancing weights [14] to post-stratify the final sample data to fit the demographic margins.

### 2.2. Measurements

As our dependent variable, we focused on participants’ willingness to take a booster vaccine dose. Since 15 December 2021, we have thus asked participants in our survey the following question: “Have you received a third vaccine dose (a so-called booster jab)?” (note that we do not specify the specific vaccine, but the available vaccines in the Danish vaccination program are the vaccines developed by Pfizer/BioNTech and Moderna). Below, we present the overall distribution of responses on our dependent variable (see Table 2). Note that by 13 February, the share of the adult population that had received a booster dose was about 76.5 percent (the number of revaccinated people was retrieved from the Danish Center for Disease Control [15], whereas the effective number of inhabitants aged 18 or older was retrieved from Danish Statistics [13]). The percentage of sample respondents who reported to have received the third vaccine dose (response category 1 in Table 2) cannot be compared directly to this population share because the sample estimates are based on the entire observation period from 15 December 2021 to 13 February 2022, whereas the population benchmark was observed 13 February 2022. For the analyses, we collapsed response categories 1, 3, and 5 into overall willingness to take the booster vaccine dose (coded as 100). We coded the remaining categories (2, 4, and 6) as refusal (coded as 0). Overall, this yielded an estimated willingness to take up the booster dose of about 87 percent (95% CI: 86.3; 87.8).

Our main predictors are based on the protection motivation theory framework and include measures that reflect threat appraisals or coping appraisals, respectively. In particular, we used validated measures of personal threat appraisal, societal threat appraisal, self-efficacy, response efficacy, and response cost [7,8,11,12,16]. In line with previous literature, we distinguished between the perceived threat that the coronavirus poses to the society and the perceived threat that the individual experiences when assessing threat appraisals [7,8,16]. We asked participants, “To what degree do you feel that…”. Then, they responded to two statements: one about threat to the society (“...the coronavirus is a threat to the Danish society”) and one about individual threat (“...you are threatened by the coronavirus”). Similarly, when measuring coping appraisals, we distinguished between self-efficacy, response efficacy, and response costs, in line with the literature [11,12]. For each of the three coping measures, we generated scales based on participants’ agreement with two statements. For self-efficacy, the statements were: (1) “It is easy for me to follow the advice of the health authorities” and (2) “I feel confident that I can follow the advice of the health authorities if I want to”. The two statements related to response efficacy read: (1) “If I follow the advice of the health authorities, I will be as safe as possible during the corona epidemic” and (2) “If I follow the advice of the health authorities, I will help protect others from the coronavirus”. Finally, another two statements related to response costs were: (1) “If I follow the advice of the health authorities, my relationship with people outside the family will be impaired” and (2) “If I follow the advice of the health authorities, my life will be degraded”.

For all of the protection motivation items, participants responded on seven-point scales ranging from 1 (completely disagree) to 7 (completely agree). We excluded “do not know” responses. We treated all of the protection motivation measures as continuous variables and centered them on their mean with a standard deviation of 0.5 (this means that a one-unit change in these measures equals a change of two standard deviations on the original scales). In the models below, we also included participants’ sex (whether they define themselves as male or female) and their age (measured in six age intervals, including 18–29, 30–39, 40–49, 50–59, 60–69, and 70+ years old) as additional predictors.

### 2.3. Statistical Analysis

In all analyses, we model booster vaccine willingness using OLS (ordinary least squares) regression. That is, we regressed our dependent variable—booster vaccine willingness, as described above—on our set of predictors, including the protection motivation variables (individual threat, societal threat, self-efficacy, response efficacy, and response cost) and the demographic variables (sex and age). When estimating the association between willingness to take the booster vaccine and each of the protection motivation scales, this means that the coefficients can be interpreted as the percentage points change in booster vaccine willingness associated with a two-standard-deviation change in the respective predictor (given the scaling of the variables). The demographic variables were entered into the models as indicator variables. This means that the estimated coefficients can be interpreted as the percentage points change in booster vaccine willingness compared to the reference category (for age: 18–29 years; for sex: male). In all analyses, we post-stratified the results using the weights described above. We employed robust standard errors in all estimations to account for heteroscedasticity. In the estimations, we made use of two samples. First, we focused our estimations on participants who had taken up primary vaccination. In particular, we utilize that we also asked participants about primary vaccine uptake to limit the sample to participants that responded that they either are (1) fully vaccinated (received two doses), (2) partially vaccinated (received one dose), or (3) wish to be vaccinated but had previously refused an invitation (we labeled this the “primary vaccination sample”). Second, we reran the estimations in the full sample (we labeled this the “full sample”), including participants who indicated that they would not take the primary vaccines.

## 3. Results

We estimate that about 87 percent (95% CI: 86.3; 87.8) among the adult Danish population would willingly take a booster vaccine. This compares to an estimated 90.5 percent in our sample of adult Danes who responded that they are either (1) fully vaccinated (received two doses), (2) partially vaccinated (received one dose), or (3) wish to be vaccinated but has previously refused an invitation. When estimating booster vaccine willingness among participants who reported to be willing to take the primary vaccine, we obtained an estimated booster vaccine willingness of about 95.5 percent (95% CI: 95.2; 95.8). In other words, this implies that the assumption that those who accepted the primary vaccination will also automatically accept a booster dose does not seem valid. Specifically, we estimate that about 4.5 percent of those who accepted primary vaccination will refuse to get the booster dose.

In Figure 2, we plot the estimated predicted booster vaccine willingness from bivariate regressions between willingness, age, and sex. The top panels of the figure are based on the primary vaccination sample, whereas the bottom panels are based on the full sample. Across both samples, we observe a clear age pattern. The younger the participant, the less likely that she is willing to take the booster dose. Focusing on the primary vaccination sample (those who would be invited for a booster jab), we observe that the estimated willingness to take the booster is 91.9 percent (95% CI: 90.8; 93.1) among respondents aged 18–29, 88.9 percent (95% CI: 87.3; 90.5) among those aged 30–39, 95.2 percent (95% CI: 94.1; 96.32) among those aged 40–49, 97.9 percent (95% CI: 97.4; 98.4) among those aged 50-59, 98.8 percent (95% CI: 98.3; 99.3) among those aged 60–59, and 99.3 percent (95% CI: 98.8; 99.9) among respondents aged 70+. Unsurprisingly, Figure 2 shows that this age pattern is more pronounced in the full sample that includes participants who are not willing to take primary vaccination. Unlike the clear age pattern, we observe no statistically meaningful differences in willingness across the sexes in the primary vaccination sample. On the contrary, in the full sample, we find that females are statistically significantly more likely to take the booster vaccine by 3.3 percentage points (95% CI: 1.8; 4.8). This reflects that females in the sample display a higher likelihood of being willing to take the primary vaccine.

In Table 3, we model booster vaccine willingness using multivariate OLS regression while distinguishing between the two samples as specified above. Specifically, we regressed willingness on our set of protection motivation predictors and the demographics. Focusing on the sample that willingly accepted primary vaccination, Table 3 confirms the demographic patterns discussed above. There is a clear difference between old and young, with the elderly being more likely to accept the booster dose, whereas there is no heterogeneity across the sexes. Moving to the protection motivation predictors, we observe that four out of five predictors correlate statistically significantly with booster willingness. Thus, a two-standard-deviation increase in societal threat is associated with a 2.75 percentage point (95% CI: 1.98; 3.52) increase in booster willingness. A two-standard-deviation increase in self-efficacy is associated with a 1.39 percentage point (95% CI: 0.33; 2.44) increase in booster willingness. A two-standard-deviation increase in response efficacy is associated with a 4.96 percentage point (95% CI: 3.92; 6.00) increase in booster willingness. A two-standard-deviation increase in response cost is associated with a −2.62 percentage point (95% CI: −3.47; −1.79) decrease in booster willingness. Turning our attention to the full sample estimates, we observe a similar but stronger empirical pattern. This is due to the fact that the correlates are strong predictors of primary vaccination willingness [9,10].

In Table 4, we rerun the results from the first column of Table 3 (the primary vaccination sample) while splitting the sample into young (18–39 years), medium-aged (40–59 years), and old (60+ years). The results from these subgroup analyses show that there is substantial variation in the correlations between protection motivation and booster vaccine willingness across age groups. On response efficacy, for example, we observe that a two-standard-deviation increase in response efficacy is associated with a 10.69 percentage point (95% CI: 7.99; 13.38) increase in booster willingness among the young, an increase of 3.65 percentage points (95% CI: 2.39; 4.91) among the medium-aged, and a 1.47 percentage point (95% CI: 0.57; 2.37) increase among the old. Similar patterns are found for the remaining protection motivation factors, with the strongest associations observed among the young, more moderate associations among the medium-aged, and the weakest associations observed among the old. In fact, for the old, all associations—except for the positive response efficacy correlation—remain statistically indistinguishable from zero at conventional levels of statistical significance.

The results above do not only show that the associations between booster vaccine willingness and protection motivation vary substantively by age; they also offer a potential explanation for the age differences that we observed above. Thus, one interpretation of the empirical patterns above is that irrespective of their motivation to engage in protective behavior, the elderly are prepared to take up boosters. On the contrary, the willingness of the young depends on their sense of the society being threatened, their sense of self-efficacy, their sense of response efficacy (i.e., that their behavior makes a difference), and their perceptions of costs from complying with the recommended response. To illustrate this point, Figure 3 plots the predicted probabilities of taking boosters as a function of each of the protection motivation factors while splitting the sample into young (solid lines), medium-aged (dash-dotted lines), and old (dashed lines). The figure corroborates the above interpretation. The old are willing to take up boosters at a high level, irrespective of their protection motivation, whereas the willingness of the young depends on their motivation to engage in protective behavior.

## 4. Discussion

Given the rapid spread of the Omicron variant, a key aspect of epidemic control is populations’ willingness to receive COVID-19 boosters. On the basis of high-quality data from Denmark, we estimate an overall willingness in the adult Danish population of about 87 % and a willingness of 95.5 % among primary vaccine takers. However, the results also demonstrate that these percentages are significantly lower among younger populations, as well as among groups who do not see COVID-19 as a threat towards society (‘societal threat’), those who do not feel that they have the ability to follow recommendations (‘self-efficacy’), those who do not perceive the advice of the health authorities as effective against disease spread (‘response efficacy’), and those who feel that the costs of following recommendations are high (‘response cost’). On the contrary, we did not find a difference in booster vaccine willingness across the sexes, and we did not observe a correlation between booster vaccine willingness and perceived personal threat among those who had taken primary vaccination.

The results of people’s motivation to engage in protective behavior largely follow previously reported results in the literature on vaccine intentions in general. Such literature, for example, shows that protection motivation predicts people’s intention to receive a seasonal influenza vaccination [17] and follow MMR vaccination recommendations [18]. More specifically in this context, previous literature shows that protection motivation is important for understanding people’s intention to receive a primary vaccination against COVID-19 [9,10,19,20]. To the best of our knowledge, we are the first to demonstrate the usefulness of the protection motivation framework for understanding people’s willingness to take COVID-19 boosters.

The finding that younger individuals are more hesitant to receive the booster dose is consistent with previous literature from various countries [21,22,23,24,25], as well as studies of age patterns in primary vaccination against COVID [26,27,28,29,30,31]. Only a few studies do not replicate this age pattern when estimating the willingness to receive a booster dose [32,33]. Our subgroup analyses, wherein we reran the analyses while splitting the sample into young, medium-aged, and old, provide an explanation for the observed age pattern: the old are willing to take boosters at a high level, irrespective of their protection motivation, whereas the willingness of the young depends on their sense of society being threatened, their sense of self-efficacy, their sense that the proposed behavior makes a difference (i.e., response efficacy), and their perceptions of cost related to complying with the recommended response (i.e., response cost).

The finding that there is no significant difference between men and women in booster willingness among those having received a primary vaccine is consistent with another Danish study [21]. However, other studies provide more mixed evidence. Some studies show that willingness to receive a booster vaccine is higher among men [23,24], whereas others report that willingness is higher among women [22].

For epidemic surveillance, including when doing formal modelling of the epidemic trajectory, it is important to consider the age dependency of the willingness to receive booster vaccinations, even among those already vaccinated. For epidemic control, it is key for health communicators to engage in communication that addresses the concerns of those who are hesitant. In particular, there is a need to address why the spread of Omicron poses a threat to society and how booster vaccinations will enable societies to cope, as the perception of a societal treat and perceived response efficacy among individuals are closely related to booster vaccine uptake among individuals.

## 5. Conclusions

Health authorities can assume high willingness to receive a booster vaccination among those already vaccinated, but willingness decreases among younger groups, as well as groups with lower concern and trust in authorities.

## Figures and Tables

**Figure 1 vaccines-10-00425-f001:**
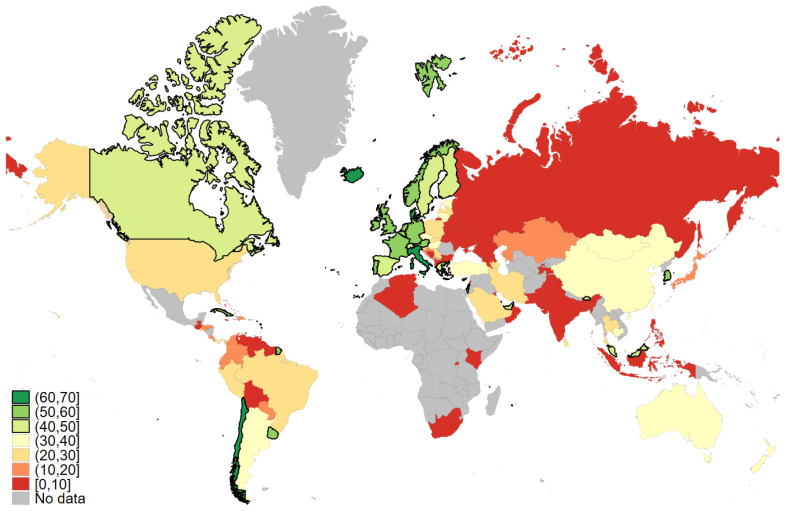
COVID-19 vaccine boosters administered per 100 people. Note: Data retrieved from *Our World in Data*, 14 February 2022.

**Figure 2 vaccines-10-00425-f002:**
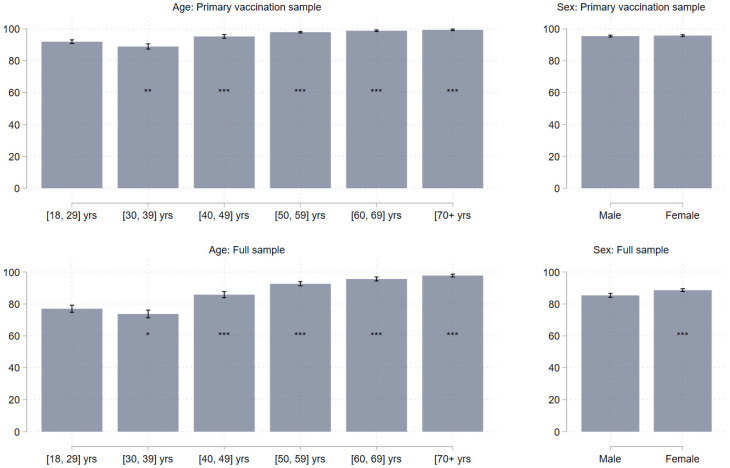
Booster vaccine willingness by age and sex. Note: Bars reflect the estimated willingness to take a booster dose. Whiskers denote 95% confidence intervals. * *p* < 0.05, ** *p* < 0.01, *** *p* < 0.001. Stars reflect whether there is a statistically significant difference between the reference category (age: 18–29 yrs; sex: male) and each of the other categories. Robust standard errors. Top panels reflect estimates from the primary vaccination sample (30,653), whereas the bottom panels reflect estimates from the full sample (*N* = 31,721).

**Figure 3 vaccines-10-00425-f003:**
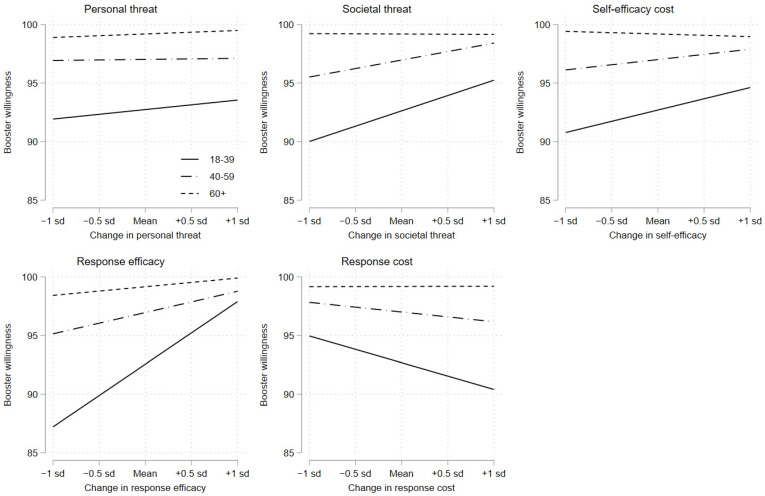
Associations between protection motivation and booster willingness by age. Note: Changes in the predicted willingness to take up boosters over personal threat, societal threat, self-efficacy, response efficacy, and response cost. Estimates are based on OLS regression. Solid lines are estimated probabilities for the young (18–39 years), dash-dotted lines for the medium-aged (40–59 years), and dashed lines for the old (60+ years).

**Table 1 vaccines-10-00425-t001:** Population margins versus sample and weighted sample margins.

	Sample	Weighted Sample	Population Benchmark
Vaccination status			
Fully vaccinated (2 jabs)	96.0%	89.1%	89.1%
Partially vaccinated (1 jab)	0.5%	1.2%	1.2%
Unvaccinated	3.5%	9.7%	9.7%
Education			
Primary school	9.9%	25.2%	25.1%
Vocational	23.8%	29.5%	29.5%
High school	10.8%	10.9%	10.9%
Bachelor’s degree	36.2%	23.0%	23.0%
Master’s degree	19.3%	11.4%	11.4%
Sex x age			
Female 18–29 years	9.9%	9.7%	9.7%
Female 30–39 years	6.8%	7.2%	7.2%
Female 40–49 years	9.8%	8.1%	8.1%
Female 50–59 years	9.5%	8.5%	8.5%
Female 60–69 years	8.5%	7.2%	7.2%
Female 70+ year	7.5%	9.9%	9.9%
Male 18–29 years	8.5%	10.1%	10.1%
Male 30–39 years	6.9%	7.5%	7.5%
Male 40–49 years	9.3%	8.1%	8.1%
Male 50–59 years	8.5%	8.6%	8.6%
Male 60–69 years	8.3%	7.0%	7.0%
Male 70+ year	6.6%	8.1%	8.1%
Region			
Capitol	31.5%	31.8%	31.7%
Midtjylland	23.7%	22.6%	22.6%
Nordjylland	9.8%	10.2%	10.2%
Sjælland	13.6%	14.4%	14.4%
Syddanmark	21.4%	21.0%	21.0%
Party			
Socialdemokratiet	23.46%	21.68%	21.68%
Radikale Venstre	6.75%	7.21%	7.22%
Det Konservative Folkeparti	8.11%	5.54%	5.54%
Nye Borgerlige	1.63%	1.97%	1.97%
Klaus Riskær Pedersen	0.20%	0.70%	0.70%
Socialistisk Folkeparti	6.36%	6.45%	6.45%
Liberal Alliance	1.74%	1.95%	1.95%
Kristendemokraterne	0.84%	1. 44%	1.44%
Dansk Folkeparti	3.08%	7.31%	7.31%
Stram Kurs	0.14%	1.50%	1.50%
Venstre	14.88%	19.58%	19.58%
Enhedslisten	6.48%	5.81%	5.81%
Alternativet	1.31%	2.47%	2.47%
Other	25.03%	16.38%	16.37%

Note: The fourth column (population benchmark) reports the population margins with respect to each of the characteristics in the rows of the table, including vaccination status, sex, age, region of residence, and party choice at the last first-order national election. The second (sample) and third (weighted sample) columns of the table report the same margins for the sample that we utilized in this study for the unweighted and weighted data, respectively.

**Table 2 vaccines-10-00425-t002:** Distribution of responses.

Response Category	Response Label	Percent	95% CI
1	I have received the 3rd vaccine dose	71.05	70.03; 71.86
2	I have received an invitation to the 3rd vaccination dose, but a do not wish the 3rd dose	2.75	2.43; 3.07
3	I have not yet received the invitation to the 3rd vaccination dose, but I wish to be vaccinated with the 3rd dose	7.86	7.49; 8.22
4	I have not yet received the invitation to the 3rd vaccination dose, and I do not wish to be vaccinated with the 3rd dose	7.08	6.45; 7.71
5	I have received the invitation to the 3rd vaccination dose, and I wish to be vaccinated, but have not yet been vaccinated with the 3rd dose	8.11	7.72; 8.51
6	Do not want to answer	3.15	2.74; 3.56

Note: Response labels correspond to the response options that participants saw in the questionnaire. Percentages and 95% confidence intervals are calculated on the basis of the weighted sample data.

**Table 3 vaccines-10-00425-t003:** Booster vaccine willingness, regression results.

	Sample: Primary Vaccination		Sample: Full	
	Estimate	95% CI	Estimate	95% CI
Demographics				
18–29 yrs	ref	ref	ref	ref
30–39 yrs	−3.61 **	−5.71; −1.52	−3.55 *	−6.38; −0.73
40–49 yrs	2.56 ***	0.91; 4.21	6.31 ***	3.69; 8.92
50–59 yrs	4.57 ***	3.29; 5.85	9.46 ***	7.15; 11.77
60–69 yrs	4.94 ***	3.70; 6.18	10.30 ***	8.03; 12.57
70+ yrs	5.10 ***	3.81; 6.40	11.17 ***	8.99; 13.34
Sex: male	ref	ref	ref	ref
Sex: female	−0.26	−1.06; 0.54	0.13	−1.14; 1.41
Protection motivation				
Personal threat (2 sd)	0.80	−0.02; 1.63	2.07 **	0.73; 3.41
Societal threat (2 sd)	2.75 ***	1.98; 3.52	9.52 ***	8.14; 10.90
Self-efficacy (2 sd)	1.39 *	0.33; 2.44	4.53 ***	2.50; 6.56
Response efficacy (2 sd)	4.96 ***	3.92; 6.00	15.44 ***	13.28; 17.60
Response cost (2 sd)	−2.62 ***	−3.47; −1.79	−5.87 ***	−7.23; −4.52
Constant	93.58 ***	92.03; 95.14	83.43 ***	80.48; 86.38
R^2^	0.07		0.23	
Observations	27,402		28,257	

Note: Unstandardized (weighted) OLS regression coefficients. * *p* < 0.05, ** *p* < 0.01, *** *p* < 0.001. 95% confidence intervals. Robust standard errors.

**Table 4 vaccines-10-00425-t004:** Booster vaccine willingness estimates by age.

	Sample: 18–39 Years		Sample: 40–59 Years		Sample: 60+ Years	
	Estimate	95% CI	Estimate	95% CI	Estimate	95% CI
Demographics						
Sex: male	ref.	ref.	ref.	ref.	ref.	ref.
Sex: female	−0.45	−2.43; 1.53	−0.49	−1.66; 0.67	0.27	−0.43; 0.98
Protection motivation						
Personal threat (2 sd)	1.64	−0.16; 3.44	0.21	−0.10; 1.42	0.59	−0.44; 1.63
Societal threat (2 sd)	5.25 ***	3.15; 7.34	2.91 ***	1.90; 3.92	−0.08	−1.00; 0.84
Self-efficacy (2 sd)	3.85 **	1.00; 6.69	1.77 *	0.42; 3.12	−0.47	−1.13; 0.19
Response efficacy (2 sd)	10.69 ***	7.99; 13.38	3.65 ***	2.39; 4.91	1.47 **	0.57; 2.37
Response cost (2 sd)	−4.56 ***	−6.89; −2.24	−1.68 **	−2.71; −0.64	0.01	−0.74; 0.76
Constant	93.07 ***	90.13; 96.01	97.61 ***	95.59; 99.64	98.76 ***	97.47; 100.03
R^2^	0.07		0.04		0.01	
Observations	8755		10,484		8163	

Note: Unstandardized (weighted) OLS regression coefficients. * *p* < 0.05, ** *p* < 0.01, *** *p* < 0.001. 95% confidence intervals. Robust standard errors.

## Data Availability

The data presented in this study are openly available in OSF at https://osf.io/kg8xh/ (doi:10.17605/OSF.IO/KG8XH).

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
