# Peer review of "Willingness to Take the Booster Vaccine in a Nationally Representative Sample of Danes"

_vaccines, 2022, doi:10.3390/vaccines10030425_

Round 1
Reviewer 1 Report
This short communication describes the results of a socialdemographic study regarding the viewpoints of a representative human population in relation to a vaccination program.
This study involves a significant big number of individuaks, and of statistical expertise. The authors have already conducted similar studies.
As stated, conclusions are important for health authorities.
The text is very well written and presented, and the English usage is quite good.
Author Response
Reviewer 1:
This short communication describes the results of a socialdemographic study regarding the
viewpoints of a representative human population in relation to a vaccination program.
This study involves a significant big number of individuaks, and of statistical expertise. The authors
have already conducted similar studies.
As stated, conclusions are important for health authorities.
The text is very well written and presented, and the English usage is quite good.
We thank reviewer 1 for the kind words.

Reviewer 2 Report
Specific comments are included below:
General question: did you differentiate between the specific vaccine? For many readers, it may not be not clear what vaccine(s) are being used in Denmark. It would be interesting to see the breakdown of responses by vaccine (if those data are available).
Lines 19-20: Recommend removing "so-called" as "booster" alone is acceptable.
Lines 34-35: Please specify the current (accepted) total Danish population.
Line 37: the eBoks system is unknown to those outside of Denmark/Europe. It would be good to include a reference (if possible) or at least a few more lines about this system.
Line 46: Is Table 1 specific to the entire Danish population or just the 2,731 participants for this study? It would benefit to clarify this. In addition, the footnote for Table 1 should be immediately below the table instead of at the bottom of the page.
Line 50: Please reference Table 2
Line 79: Figure 1. Were you able to conduct statistical analyses on these data? It would be particularly interesting to see the differences, if any, in the younger age groups. Perhaps this is covered by Table 3 (please confirm).
Line 83: Please define OLS
Line 104: Table 3. It is unclear what "ref" is. Further, the R2 values are rather low; it is worth discussing this in context of your conclusions.
Final notes: this study was conducted over a short period of time at the end of 2021. Have you been able to update your data set and associated statistics? If so, it is worth discussing this. In addition, it would be good to stress that these conclusions may not apply to all regions/countries, but that indeed using methods to assess vaccine willingness/unwillingness are critical.
Author Response
Reviewer 2
Specific comments are included below:
General question: did you differentiate between the specific vaccine? For many readers, it may not
be not clear what vaccine(s) are being used in Denmark. It would be interesting to see the
breakdown of responses by vaccine (if those data are available).
The reviewer makes a good point that it could be interesting to see the breakdown of responses by
vaccine. However, we did not ask a question that allows us to distinguish between the different
vaccines. The Danish vaccine program includes only the Pfizer and Moderna vaccines as both the
Astrazeneca and Johnson & Johnson vaccines were excluded from the program after worries over
side-effects. The Astrazeneca vaccine was initially part of the vaccination program, but was
suspended on March 11, 2021. We now specify this in the manuscript.
Lines 19-20: Recommend removing "so-called" as "booster" alone is acceptable.
We follow this recommendation.
Lines 34-35: Please specify the current (accepted) total Danish population.
In the manuscript, we have now specified that the that “the total Danish population aged 18 or older
amount to 4,721,691 by the first quarter of 2020 according to data retrieved from Danish Statistics”
Line 37: the eBoks system is unknown to those outside of Denmark/Europe. It would be good to
include a reference (if possible) or at least a few more lines about this system.
This is a good point. We have now included the following description in the manuscript:
Participants are invited to take the survey via eBoks. eBoks is the official electronic
mail system used by the authorities and other institutions to communicate with
citizens. eBoks is linked to an individual’s social security number and follows the him
or her the entire life. This link between the social security number and eBoks is the
reason that we are able to utilize this electronic resource as a means of sampling.
Line 46: Is Table 1 specific to the entire Danish population or just the 2,731 participants for this
study? It would benefit to clarify this. In addition, the footnote for Table 1 should be immediately
below the table instead of at the bottom of the page.
The “population benchmark” in the fourth column of Table 1 refers to the population margins on
each of the characteristics that are shown in the rows of Table 1. The “sample” (second column) and
“weighted sample” (third column) columns of Table 1 report the estimated margins for the same
characteristics in the sample we use in this study – i.e., the 2,731 (or now 31,721 upon updating the
data) participants. In the text we have now specified it more clearly that this study builds on the
smaller sample. More specifically we have included the following sentence:
“Note, that data for this study includes a subset of participants (N = XXXX) as we
focus on booster vaccination willingness that we only started measuring from
December 15, 2021 (cf. below).”
The point is that when we compare the sample margins to the population margins, then it shows that
the data is of high quality in the sense that it quite closely reflects the underlying Danish population
(at least with respect to the reported characteristics).
The footnote was actually not meant for Table 1. We have now added a note for Table 1, instead,
that specifies for points made above.
Line 50: Please reference Table 2
We thank the reviewer for noticing this issue and have now corrected it.
More generally, we have now added a separate measurement section where we discuss the
measurement of the dependent variable (and reference Table 2) as well as detailing the independent
variables.
Line 79: Figure 1. Were you able to conduct statistical analyses on these data? It would be
particularly interesting to see the differences, if any, in the younger age groups. Perhaps this is
covered by Table 3 (please confirm).
We have detailed that the results in Figure 1 (now Figure 2) are based on bivariate regressions. As
per suggestion of reviewer 3, we have included “stars” into the figure to indicate significant
differences between each group and the reference category.
We follow the reviewer’s advice and discuss in more detail differences between groups. The
reviewer is also correct in that the differences between groups are also covered in Table 3 (the
difference between Table 3 and Figure 2 is however that whereas Figure 2 is based on bivariate
regressions Table 3 is multivariate).
Line 83: Please define OLS
We specified that OLS means “ordinary least squares”
Line 104: Table 3. It is unclear what "ref" is. Further, the R2 values are rather low; it is worth
discussing this in context of your conclusions.
By “ref” we mean reference categories. Given that the age and sex variables are categorical
variables, then each entry in the cells of the table on these variables “refer” to a reference category.
That means that the estimated coefficient gives the difference between the specific category and the
reference. For age, the reference category is the young (18-29 years old). For sex, the reference is
male.
Final notes: this study was conducted over a short period of time at the end of 2021. Have you
been able to update your data set and associated statistics? If so, it is worth discussing this. In

Reviewer 3 Report
Although the article is a short communication, it is still too brief compared to other communications in this journal. All sections should be improved with more information.
- Introduction: the introduction should be expanded more. For example, what is the booster willingness overall in Denmark or other countries?
- Methods: what statistical methods were used for data in Table 3? I know that you used OLS regression from other sections, but all methods should be explained in this section.
- Figure 1: change the y-axis to percentage. Max is 100%, instead of 1. Increase the font size of categories on the x-axis (e.g., the labels for the categories such as "18-29 yrs", "male" etc are too small)
- Figure 1 and Table 3: Combine Figure 1 and demographics in Table 3. Mark the categories with a significant difference and put an explanation (e.g., Significant compared to subjects aged 18-29 year old/ Statistical significance compared to male at p-value < 0.05)) in the footnote.
- Leave categories of Protection Motivation and their data only in Table 3.
- What does it mean 2 sd and what do the estimates mean in protection motivation categories? Any statistical significance? Mark the group or category with a statistical significance.
- Table 3: societal treat: 0.02-0.07 was in a different format from other categories.
- What does R2 mean in Table 3?
- Discussion: Expand it more with more references. 1) Are your findings similar or different from other countries? What are the possible explanations for your results? For example, your results showed the willingness is lower in younger generations. Why? What is the comparison to the primary vaccination? There could be some earlier studies in Denmark or other countries about the willingness of primary vaccination. Any difference or similar pattern? etc.
Author Response
Reviewer 3
Although the article is a short communication, it is still too brief compared to other communications
in this journal. All sections should be improved with more information.
The reviewer made a fair point that the communication was originally too short compared with
other communications in Vaccines. Accordingly, we have strengthen each section of the
manuscript. Please see responses below for details.
1. Introduction: the introduction should be expanded more. For example, what is the booster
willingness overall in Denmark or other countries?
We agree with the reviewer that we could have done more in the original manuscript to
frame the contribution of the study more clearly. Therefore, we have now contextualized
COVID-19 booster uptake in the introduction. In particular, we have added a new figure
(Figure 1) that shows the shows COVID-19 vaccine boosters administered per 100 people
across the world. With about 62 percent of the total population having received a booster,
Denmark is only surpassed by Chile (about 69 percent) in re-vaccinating their population
with a third dose. We have, moreover, added a paragraph in the introduction that details why
we focus on factors related to protection motivation theory for understanding booster
vaccine willingness.
2. Methods: what statistical methods were used for data in Table 3? I know that you used OLS
regression from other sections, but all methods should be explained in this section.
To first answer the specific questions, yes we used OLS regression in Table 3.
That said, then we fully agree with the reviewer that the methods section should have been
more detailed in the original manuscript. Accordingly, we have now expanded the methods
section substantially devoting separate sections for data collection, measurements, and
statistical analysis.
3. Figure 1: change the y-axis to percentage. Max is 100%, instead of 1. Increase the font size
of categories on the x-axis (e.g., the labels for the categories such as "18-29 yrs", "male" etc
are too small)
We follow the recommendations of reviewer 3. And we also use the 0-100 variable instead
of 0-1 for the results in the regression table.
4. Figure 1 and Table 3: Combine Figure 1 and demographics in Table 3. Mark the categories
with a significant difference and put an explanation (e.g., Significant compared to subjects
aged 18-29 year old/ Statistical significance compared to male at p-value < 0.05)) in the
footnote.
We follow the recommendations made by the reviewer.
5. Leave categories of Protection Motivation and their data only in Table 3.
As explained below in comment (6) each of the protection motivation variables are
continuous scales and treated accordingly. The estimated associations between
willingness to take up the booster vaccine and each of these scales, this means that the
coefficients can be interpreted as the change in booster vaccine willingness associated
with a 2 standard deviation change in the respective predictor”.
6. What does it mean 2 sd and what do the estimates mean in protection motivation categories?
Any statistical significance? Mark the group or category with a statistical significance.
In the measurement section, we have now clearly explained the measurement of all the
protection motivation theory variables and that they can be regarded as scales. “2 sd” simply
refers to the re-scaling of these scales. We now detail clearly in the measurement section
that:
“We treat all the PMT variables as continuous variables and center them on their mean
with a standard deviation of 0.5.”
This means that a 1-unit increase on each variable simply amounts to a 2 standard deviation
change on the variable (something which is often used as a “typical change”/or substantively
interesting change within econometrics). We explain the implications of this scaling in the
manuscript:
“When estimating the association between willingness to take up the booster vaccine
and each of these scales, this means that the coefficients can be interpreted as the
change in booster vaccine willingness associated with a 2 standard deviation change in
the respective predictor”.
7. Table 3: societal treat: 0.02-0.07 was in a different format from other categories.
Thank you for noticing. This was a mistake and changed accordingly.
8. What does R2 mean in Table 3?
It means R-squared. A measure of the explained variance, which is routinely reported for
OLS regression models.
9. Discussion: Expand it more with more references. 1) Are your findings similar or different
from other countries? What are the possible explanations for your results? For example,
your results showed the willingness is lower in younger generations. Why? What is the
comparison to the primary vaccination? There could be some earlier studies in Denmark or
other countries about the willingness of primary vaccination. Any difference or similar
pattern? etc.
We agree with the reviewer that we could have done more to make our contribution clearer
and relate to previous literature in the Discussion. We have tried to do so now. Specifically
addressing previous literature on protection motivation theory and vaccine willingness both
in general and in the specific context of COVID-19 vaccines.

Round 2
Reviewer 3 Report
The authors improved the manuscripts as the reviewers suggested. But it will be much more informative to explore the reason for the disparities by age group if the authors conduct a further minor revision.
- It would be good to add the trend found in other studies in other countries. Are there any other published papers about the disparities of the willingness to have a booster shot by age group, gender, and other factors? Are your findings consistent with them or unique? It doesn't need to be long but just 2-3 sentences with references.
- It still needs a brief discussion why young generation is worse than older generations. Authors include protection motivation factors in the study. It would be helpful to see which protection motivation factors are different by age group and it can be the explanation of the disparities of booster willingness by each group. Figure 2 already shows the difference by age group and gender, and Table 3 can be modified. Authors can show the estimates of protection motivation factors by each age group and add several sentences in results and discussion.
- For example, in ages 18-39, the societal threat is a significant factor, but the personal threat doesn't change the willingness. Then, we can discuss this factor as the young generation consider them not a risk group. But in ages 70+, response cost doesn't affect the difference, but the personal treat is a significant factor. It could be because they consider themselves as a high-risk group and they want to get a booster shot to protect themselves.
Author Response
The authors improved the manuscripts as the reviewers suggested. But, it will be much more informative to explore the reason for the disparities by age group if the authors conduct a further minor revision.
In the original draft, we did not focus too much on the age heterogeneity in booster willingness. However, the reviewer makes a good point that this is essential for the manuscript. Accordingly, we follow all of his or her advices (please see below).
- It would be good to add the trend found in other studies in other countries. Are there any other published papers about the disparities of the willingness to have a booster shot by age group, gender, and other factors? Are your findings consistent with them or unique? It doesn't need to be long but just 2-3 sentences with references.
We believe that the reviewer is correct that the original draft could have benefitted from a discussion of other booster vaccine studies. We have accordingly added two paragraphs in the discussion section to this end.
- It still needs a brief discussion why young generation is worse than older generations. Authors include protection motivation factors in the study. It would be helpful to see which protection motivation factors are different by age group and it can be the explanation of the disparities of booster willingness by each group. Figure 2 already shows the difference by age group and gender, and Table 3 can be modified. Authors can show the estimates of protection motivation factors by each age group and add several sentences in results and discussion.
- For example, in ages 18-39, the societal threat is a significant factor, but the personal threat doesn't change the willingness. Then, we can discuss this factor as the young generation consider them not a risk group. But in ages 70+, response cost doesn't affect the difference, but the personal treat is a significant factor. It could be because they consider themselves as a high-risk group and they want to get a booster shot to protect themselves.
We follow the reviewer’s advice and split the estimated correlation by age, including young (18-39), medium-aged (40-59), and old (60+) to explore this heterogeneity. The findings show a very clear age pattern in the PMT correlations (please see the new Table 4). Across all protection motivation factors, we observe the strongest associations with booster willingness among the young, more moderate associations among the medium-aged, and the weakest associations observed among the old (in fact for most factors the associations are statistically insignificant for the old).
Overall, we take this pattern to suggest the following explanation for the age heterogeneity: Irrespective of their motivation to engage in protective behavior, the elderly will be prepared to take up boosters. To the contrary, the willingness of the young depends on their sense of the society being threatened, their sense of self-efficacy, their sense of response efficacy (i.e., that their behavior makes a difference), and their perceptions of cost related to that response. This interpretation is corroborated by Figure 3 that shows the predicted probabilities of accepting boosters across each of the PMT factors while splitting the sample by age group.